# Beyond GWAS—Could Genetic Differentiation within the Allograft Rejection Pathway Shape Natural Immunity to COVID-19?

**DOI:** 10.3390/ijms23116272

**Published:** 2022-06-03

**Authors:** Joanna Szyda, Paula Dobosz, Joanna Stojak, Mateusz Sypniewski, Tomasz Suchocki, Krzysztof Kotlarz, Magdalena Mroczek, Maria Stępień, Dawid Słomian, Sławomir Butkiewicz, Paweł Sztromwasser, Jakub Liu, Zbigniew J. Król

**Affiliations:** 1Biostatistics Group, Department of Genetics, Wrocław University of Environmental and Life Sciences, 51-631 Wroclaw, Poland; tomasz.suchocki@upwr.edu.pl (T.S.); krzysztof.kotlarz@upwr.edu.pl (K.K.); jakub.liu@gmail.com (J.L.); 2Department of Cattle Breeding, National Research Institute of Animal Production, Krakowska 1, 32-083 Balice, Poland; dawid.slomian@iz.edu.pl; 3Central Clinical Hospital of Ministry of the Interior and Administration in Warsaw, 02-507 Warsaw, Poland; paula.dobosz@cskmswia.gov.pl (P.D.); slawomir.butkiewicz@cskmswia.gov.pl (S.B.); zbigniew.j.krol@cskmswia.gov.pl (Z.J.K.); 4Department of Haematology, Transplantation and Internal Medicine, University Clinical Centre of the Medical University of Warsaw, 02-091 Warsaw, Poland; 5MNM Diagnostics, Małe Garbary 9, 61-756 Poznan, Poland; mateusz.sypniewski@mnm.bio (M.S.); pawel.sztromwasser@mnm.bio (P.S.); 6Department of Neurology and Neurophysiology, University of Zurich, Balgrist University Hospital, 8008 Zurich, Switzerland; m.mroczek888@gmail.com; 7Department of Infectious Diseases, Medical University of Lublin, 20-059 Lublin, Poland; mmaria.stepien@gmail.com

**Keywords:** KEGG pathways, allograft rejection, GWAS, COVID-19 infection, susceptibility, resistance, genetic variants, single nucleotide polymorphism, whole genome sequencing, immunisation

## Abstract

COVID-19 infections pose a serious global health concern so it is crucial to identify the biomarkers for the susceptibility to and resistance against this disease that could help in a rapid risk assessment and reliable decisions being made on patients’ treatment and their potential hospitalisation. Several studies investigated the factors associated with severe COVID-19 outcomes that can be either environmental, population based, or genetic. It was demonstrated that the genetics of the host plays an important role in the various immune responses and, therefore, there are different clinical presentations of COVID-19 infection. In this study, we aimed to use variant descriptive statistics from GWAS (Genome-Wide Association Study) and variant genomic annotations to identify metabolic pathways that are associated with a severe COVID-19 infection as well as pathways related to resistance to COVID-19. For this purpose, we applied a custom-designed mixed linear model implemented into custom-written software. Our analysis of more than 12.5 million SNPs did not indicate any pathway that was significant for a severe COVID-19 infection. However, the Allograft rejection pathway (hsa05330) was significant (*p* = 0.01087) for resistance to the infection. The majority of the 27 SNP marking genes constituting the Allograft rejection pathway were located on chromosome 6 (19 SNPs) and the remainder were mapped to chromosomes 2, 3, 10, 12, 20, and X. This pathway comprises several immune system components crucial for the self versus non-self recognition, but also the components of antiviral immunity. Our study demonstrated that not only single variants are important for resistance to COVID-19, but also the cumulative impact of several SNPs within the same pathway matters.

## 1. Introduction

COVID-19 infections pose a serious global health concern, so it remains crucial to identify the biomarkers for the susceptibility to and resistance against this disease that could help in a rapid risk assessment and reliable decisions being made on patients’ treatment and their potential hospitalisation. The knowledge of genomic variants that are associated with the susceptibility to and resistance against COVID-19 infections is essential for understanding the metabolic pathways and regulatory factors related to different presentations of the disease and, in consequence, for developing new drug targets [1].

Several studies investigated factors associated with severe COVID-19 outcomes that can be either environmental, population based, or genetic. For instance, a geographic and population variation in the disease course and outcome has demonstrated that individuals representing Black, Hispanic, or Asian ethnicities have a higher risk of death compared to Caucasians [2,3]. The identified risk factors of COVID-19 infection are advanced age, male sex, and comorbidities (especially: renal disease, oncological pathologies, chronic respiratory disease, and cardiovascular disease—excluding hypertension and dementia), but they do not fully explain the wide spectrum of disease manifestations [2,4,5,6]. Moreover, the viral load can play a role in the COVID-19 severity, as demonstrated for mortality, especially in combination with advanced age [7,8]. On the other hand, many studies have supported the hypothesis that there are patients who have a natural resistance to COVID-19 infection. For instance, children or adolescents were barely affected [9] and they showed antibody responses up to four months after infection [10]. Thus, some children appear to be able to either repel the infection without the need to strongly engage adaptive immunity or are resistant to infection.

It was also demonstrated that the genetics of the host plays an important role in the various immune responses and, therefore, in the different clinical presentations of COVID-19 infection. Risk and protective variants were associated with multiple loci on chromosomes 3, 9, 6, 12, 19, and 21 [11,12] including a segment on chromosome 9 that defines the ABO blood groups [13], a segment on chromosome 12 that contains *OAS1*, *OAS2*, and *OAS3* [14], and an isoform p41 of *CD74*, which is a part of the MHC-II that blocks the endosomal entry pathway of the virus [15]. Further significant genes were *ACE2* located on chromosome X and human leukocyte antigens (HLA).

The main aim of our study was to use variant descriptive statistics from GWAS and variant genomic annotations to identify metabolic pathways that are associated with a severe COVID-19 infection as well as pathways related to resistance to COVID-19. For this purpose, we applied a custom-designed mixed linear model implemented in custom-written software.

## 2. Results

The raw number of SNPs amounted to 43,469,928; out of these, after filtering, 9,767,423 SNPs remained for GWAS analysis (Figure 1). The most significant SNP from GWAS was located on chromosome 7 (*p* = 7.10547 × 10^−9^) within the intron of *RAPGEF5* gene (ENSG00000136237); however, this gene is not assigned to any KEGG pathway. In total, 12,736 SNPs were included in the pathway model. Out of the 288 KEGG pathway effects predicted, none were significant in the SEVERE analysis, while the Allograft rejection pathway (hsa05330) was significant (*p* = 0.01087) in the RESISTANT analysis. All 27 SNPs, each representing one gene constituting hsa05330, were significant in GWAS (Table 1, Figure 2). The majority of them were located on chromosome 6 (19 SNPs) and the remainder were mapped to chromosomes 2, 3, 10, 12, 20, and X. The summary of read depth quality underlying each variant in each sample is summarised in Appendix A. Interestingly, the comparison of *p*-values of all SNPs from GWAS with *p*-values of the subset of SNPs included in the KEGG model showed that genic SNPs have lower *p*-values than intergenic SNPs (Figure 3).

## 3. Discussion

An important property of the human immune system is natural resistance, defined as the inability of pathogens, viruses included, to infect the host or as an ability of the host to limit the disease burden after infection [16]. Two mechanisms of natural resistance have been suggested [17]. The first mechanism comprises genetic variants in genes important for entry mechanisms, especially in the receptors. The second mechanism is the rapid elimination of the pathogen by host resistance mechanisms. As for the host’s survival, the balance between resistance mechanisms aiming to eliminate host pathogens and tolerance mechanisms aiming to avoid collateral damage induced by inflammation is crucial; natural resistance is mostly related to the immune system, either to its robustness or on the genetic level [17].

In our study, we used GWAS and custom-designed models to identify molecular pathways related to different presentations of COVID-19 infections and to describe significant genetic variants within these pathways. Interestingly, the comparison between the full dataset (all SNPs analysed in GWAS) and the subset used in the estimation of KEGG pathway effects, demonstrated that the majority of most significant *p*-values were assigned to intergenic SNPs (Figure 3), which can only be used as markers for a potential association with genes, but not as causal mutations. Moreover, although the most common cause of such a significant association is linkage, it is not exclusive proof that an associated SNP always points to the closest located gene as the functional cause of a phenotype (see, e.g., [18]). It was demonstrated that SNP density across the human genome is reduced in conserved regions [19]. However, both exonic and intronic regions are constrained to the same degree and they contain a reduced density of SNPs in comparison to intergenic regions [19]. Therefore, the custom-designed models for the KEGG pathways’ analysis used in this study allowed us to consider additional aspects of the genetic background in COVID-19 infections that were not identified in single SNP GWAS analyses.

Our analysis did not indicate any significant pathway in the case of a severe COVID-19 phenotype. However, in the case of a resistant phenotype, the Allograft rejection pathway (hsa05330) was identified as significant, with 27 SNPs marking genes constituting this pathway. The Allograft rejection pathway (Figure 4), associated with resistance to COVID-19 infection, comprises several immune system components crucial for self versus non-self recognition, but also the components of antiviral immunity [20]. Molecules involved in this pathway play pivotal roles in many other physiological and pathological processes, including allergies or even an anticancer response [20]. For all those processes the interplay between at least two cells is crucial, one of which remains the immune cell. The majority of the significant SNPs from the Allograft rejection pathway marked genes from the HLA complex (18 SNPs on chromosome 6, Table 1) that encode proteins forming the major histocompatibility complex (MHC), and several of them were already indicated as proteins protective against COVID-19 infection [21].

Considering the genomic location of the 27 SNPs from the Allograft rejection pathway (Table 1), five SNPs were identified in CD (cluster of differentiation—classification determinant of immune cells) protein-coding genes, as depicted in the table below (Table 2). CD40 is a member of the TNF superfamily, constitutively expressed on B cells and APC cells [22,23]. CD40 binds its ligand CD40L, which is transiently expressed on T cells and other non-immune cells under inflammatory conditions [24]. CD28 is expressed on T cells, providing signals required for T cell activation and survival, and CD80 and CD86 are its ligands [25,26,27,28]. The activity of CD80–CD28 complex stimulates the activation of transcription factors NF-κB, promoting IL-2 production [22,26,29,30]. CD80 is also a ligand for cytotoxic T-lymphocyte antigen 4 (CTLA-4, also known as CD152), which remains constitutively expressed on most of the T cells, for example, on Tregs, in which *CTLA-4* expression increases upon activation [25,27,29]. CD28 competes with CTLA-4 for binding to CD80 and CD86 [26,29,31]. Interestingly, CD80 and CD86 proteins may act as receptors for some adenoviruses [22]. Malfunctioning CD80 molecules are also involved in some pathological conditions, such as lupus erythematosus. *CD86* is also associated with myocarditis and gallbladder squamous cell carcinoma [22].

One SNP on chromosome 12 pointed out interferon IFN-gamma, which is a cytokine produced mostly by activated T cells and NK cells and that can induce MHC expression, which makes it an important factor in transplantation. IFN plays a dual role in COVID-19 infections, on one side aggravating the symptoms, and on the other alleviating the disease course, depending on the disease stage and types of the interferon involved and on the personal predispositions [31]. This occurred in some patients shown to be associated with higher mortality [31], whereas in others poor IFN responses were associated with a more severe disease course [32,33]. In immunosuppressed patients, IFN has a beneficial effect and was applied as an adjuvant treatment [34]. It was shown in mice models that a lack of IFNg led to the failure of microcirculation and necrosis of transplants, which suggested that IFNg modulates allogenic responses and could have a protective role in the early stages of the response to an organ allograft. On the other hand, IFNg promoted graft vessel disease at the later stages [35]. Therefore, the effect of IFNg on an organ allograft depended on the time after transplantation and graft type [36]. The pivotal role of IFNg in allograft rejection has been suggested also in humans. High rates of primary and secondary rejection after hematopoietic stem cell transplantation were reported in a cohort of eight HLA-identical paediatric patients [37]. Already at day +3 after hematopoietic stem cell transplantation in 15 children experiencing graft rejection, the levels of IFNg were significantly higher than in the control group, and IFNg has been suggested to be a marker of early graft rejection [38].

Unsurprisingly, the outcome of COVID-19 has been reported to be more severe in patients with co-existing pathologies, especially those associated with an impaired immune system [39,40]. The complex relationship existing between the immune system component, cancer, and COVID-19 brings about the possibility of continuing immune checkpoint inhibitors treatment among COVID-19 positive cancer patients, as well as the use of immunotherapy for the treatment of severe COVID-19 infection. In fact, several attempts have already been made, some of which are undoubtedly very promising [40,41,42]. It is known now that cancer patients have an increased risk of developing severe COVID-19 infection often leading to hospitalization and intensive care [43,44]. Given the fact that the SARS-CoV-2 virus might potentially trigger immune system over-reactivity in some patients, T cell exhaustion has also been observed in a subset of patients. Immune checkpoint inhibitors remain a useful tool used to modulate the immune reaction [43]. It is worth emphasizing that some COVID-19 patients exhibit lymphocytopenia and suffer from T cell exhaustion, a phenomenon very common in advanced cancer disease, which—in the case of COVID-19 infection—may lead to viral sepsis and an increased mortality rate [39,42,43]. It has been observed that in cancer patients, especially among immunocompromised individuals, treatment with the immune checkpoint inhibitors may restore their antitumoral immune response [42]. The use of immune checkpoint inhibitors in COVID-19 cases has been reviewed well in [42].

Despite the promising results, we must bear in mind the fact that severe COVID-19 might also be triggered by immune system aberrations, such as inborn errors of the interferon pathways [45]. The human immune system remains extremely complex and involves many genes and genomic elements, including those genes encoding cytokines: individuals that lack specific cytokines, such as interferons, or possessing gene alleles potentially impairing immune system functioning, can be more susceptible to certain infectious diseases, including COVID-19. In these people, immunotherapy should be administered with particular caution, including cancer patients.

The immunological synapse is an important element in the Allograft rejection pathway. The structure of the immunological synapse, i.e., the gap between the immune cell and another cell, such as antigen-presenting cells, APC, together with its complex molecular interactions and canonical, means the step-manner organisation has been known as “the bull’s eye structure”, with the central complex made of MHC-TCR (major histocompatibility complex–T cell receptor), leading to it providing the first signal [29,46,47,48,49]. The interaction between MHC and TCR controls the specificity and accuracy of the immune response, as the TCR genes undergo several complicated rearrangements similar to those known from antibody genes [46,48]. Furthermore, a central element of the synapse might be a directed secretion of soluble molecules into the synaptic cleft [46,50]. All other molecules, such as the LFA-1-ICAM adhesion complex, or checkpoint molecules, form distal rings and clusters around the MHC-TCR complex. The rings are required to modulate the response by providing secondary signals since the interaction between MHC and TCR is insufficient [46,49,51]. If no secondary signal is provided, the T cell becomes anergic or even undergoes apoptosis [49]. Summing up, at least two signals are required to activate a T cell: the first is an antigen recognition by the T cell receptor and MHC complex, the second is provided by the multitude of co-stimulatory and co-inhibitory receptors and ligands interactions, all of which are crucial in the allograft rejection pathway cell to cell interconnections [52]. Interestingly, studies have reported the relationship between anti-COVID-19 immunisation and acute corneal graft rejection, regardless of the vaccine or graft type [53,54,55,56]. Moreover, acute allograft rejection after immunisation was observed in the case of kidney and liver transplants [57]. However, a direct causative effect is hard to prove and needs further studies.

Furthermore, two SNPs were found on chromosome 10 (Table 1). The first of those SNPs was located within the Fas receptor (CD95), which plays an important role in the maintenance of immune tolerance. Fas-induced apoptosis (induced by Fas–FasL interactions) is involved in the cytotoxic activity of T cells and NK cells [58,59]. Studies conducted in mice models showed that mutations in Fas or FasL that inactivated their function led to disturbances in the immune response to infections with multiple different viruses, such as the influenza virus [60], herpes simplex viruses, mouse hepatitis virus or, mousepox virus [61]. Moreover, it was shown that Fas could play a role in the destruction of graft tissue. Although mechanisms for allograft injury remain unclear, the contribution of Fas and Fas ligand (FasL) was verified in the case of different commonly transplanted organs (e.g., liver, lungs, kidneys) that were probably targeted by FasL-expressing cytotoxic T lymphocytes [62]. In humans, it was shown in vivo that an increased sCD95 is related to the rejection in liver-transplanted recipients [63,64].

The most significant SNP from GWAS, marking the *RAPGEF5* gene, with an outstanding *p*-value of 7.10547 × 10^−9^, was not assigned to any KEGG pathway. *RAPGEF5* encodes Rap guanine nucleotide exchange factor 5 protein and belongs to the Ras family of GTPases that plays an important role in cell growth, differentiation, and malignant transformation [65]. The gene has not been associated with severe COVID infection in previous GWAS.

## 4. Material and Methods

### 4.1. Sample Collection

Blood samples were collected from 1235 individuals across Poland between April 2020 and April 2021. In our analysis, a subset of 1076 samples from unrelated individuals was used. For all individuals, basic clinical data including age, gender, BMI, and comorbidities (diabetes, hypertension, ischemic heart disease, stroke, heart failure, cancer, kidney disease, hepatitis B, chronic obstructive pulmonary disease) were collected. For some participants, additional data on genetic disorders, flu, tuberculosis, and measles vaccination status, smoking habits, as well as hepatitis C infection were ascertained. Only individuals without diagnosed severe health disorders (till the moment of sample collection), such as cancer, were qualified for this study. Within this cohort, the SEVERE group (N = 235) was composed of patients with severe, life-threatening outcomes of COVID-19 infection including respiratory insufficiency, requiring intensive medical care and artificial ventilation, and NEWS (National Early Warning Score) less than 5 [66]. The RESISTANT group (N = 306) was composed of volunteers who did not contract the disease or develop any symptoms despite being highly exposed to COVID-19. This group had multiple antibody blood-based tests conducted to confirm the lack of anti-SARS-CoV-2 antibodies.

Detailed information about the cohort, including demographic and clinical features, can be found in our previous paper by Kaja et al. [67].

### 4.2. Ethical Policy

All participants of this study provided informed consent before the collection of blood samples and clinical data. The ethical approval was granted by the Ethics Committee of the Central Clinical Hospital of the Ministry of Interior and Administration in Warsaw (decisions 41/2020 from 3 April 2020 and 125/2020 from 1 July 2020). The study complied with the 1964 Helsinki declaration and its later amendments and adhered to the highest data security standards 140 of ISO 27001 and the General Data Protection Regulation (GDPR) act.

### 4.3. Total Quality Management

The project was carried out following the Total Quality Management (TQM) methodology, which ensures the quality of results and analyzes the risk and possible difficulties of the planned methodology. TQM involves defining all critical points of the procedures: reference ranges for collected biological material, material preparation, DNA isolation, DNA concentration and quality, genome sequencing, and quality control of the data. The legal and ethical transparency of the entire project was ensured, including confidentiality, integrity, and impartiality of the data.

### 4.4. Whole Genome Sequencing

Whole genomes of 1076 unrelated participants were sequenced in this study. Four mL of K-EDTA peripheral blood from participants were collected according to the standardised Quality Management System protocol. Genomic DNA was isolated from the peripheral blood leukocytes using a QIAamp DNA Blood Mini Kit, Blood/Cell DNA Mini Kit (Syngen Biotech, Wrocław, Poland) and Xpure Blood Kit (A&A Biotechnology, Gdańsk, Poland) according to the manufacturers’ protocols. The concentration and purity of isolated DNA were measured using the NanoDropTM spectrophotometer and the quality of the DNA was evaluated using gel electrophoresis. The sequencing library was prepared by Macrogen Europe (Amsterdam, The Netherlands) using TruSeq DNA PCR-free kit (Illumina Inc., San Diego, CA, USA) and 550 bp inserts. The quality of DNA libraries was measured using 2100 Bioanalyzer, Agilent Technologies, Santa Clara, CA, USA. The Whole Genome Sequencing (WGS) was performed on the Illumina NovaSeq 6000 platform using 150 bp paired-end reads, yielding a mean depth of coverage of 35.26X in the cohort.

### 4.5. Pre-Processing of Whole Genome Sequence Data

The quality of sequenced reads was assessed using FastQC v0.11.7 (github.com/s-andrews/FastQC) and reads were subsequently mapped to the GRCh38 human reference genome using Speedseq framework v.0.1.2 [68], encompassing alignment with BWA-MEM 0.7.10 [69], using SAMBLASTER v0.1.22 [70] duplicate removal, and Sambamba v0.5.9 [71] for sorting and indexing. Mapping coverage was calculated for reads with MQ > 0 using Mosdepth v0.2.4 [71]. Single nucleotide variants in the nuclear genome were detected using DeepVariant v0.8.0 [72] and jointly genotyped with GLnexus v1.2.6-0-g4d057dc [73]. Next, multiallelic variant calls were decomposed into monoallelic and normalised using BCFtools v1.9 [74]. The raw SNP-set was edited by excluding SNPs that did not meet the following criteria: SNP call rate > 95%, *p*-value for HWE test > 0.0001, Minor Allele Frequency > 0.001, and being biallelic. Filtering was performed using PLINK 1.9 [75].

### 4.6. Phenotype Encoding

The original classification of the infection outcome was divided into five categories: control, resistant, benign, mild, and severe. However, in GWAS, binary phenotypes were defined. For the analysis of the resistance to COVID-19 infection (hereafter termed RESISTANT analysis) the resistant group was coded as “1”, the control group was removed from the analysis, and all the other groups were coded as “0”. For the analysis of the susceptibility to severe COVID-19 infection (hereafter termed SEVERE analysis) the severe group was coded as “1”, the control group was removed from the analysis, and all the other groups were coded as “0”.

### 4.7. Genome-Wide Association Study

The mixed linear model, simultaneously fitting effects of all 9,767,423 SNPs for the 1076 individuals, was applied for GWAS. In this model the genetic similarity between individuals was partitioned into a component that is due to the variation in SNP genotypes among patients and a “rest” component, that was not assessed by SNPs and thus represents a pure polygenic part of the total phenotypic variation. The model is given by
(1)$$y=\mu +X\beta +Z_{1}u+Z_{2}a+e,$$
where y (1076) represents a vector of binary phenotypes representing the SEVERE or the RESISTANT phenotype, μ (1076) is a general mean, β (3) is a vector of fixed effects represented by sex and age at data sampling with a corresponding design matrix X (1076 × 3), u (1076) is a vector of random additive polygenic effects of individuals assessed by the variation in SNP genotypes, with a corresponding design matrix $Z_{1}$ (1076 × 9,767,423), while a (1076) is a vector of the rest component of random additive polygenic variation among individuals that were not assessed by SNPs, with a corresponding design matrix $Z_{2}$ (1076 × 1076), e (1076) is a vector of residuals with $e\sim N(0,I{\hat{\sigma }}_{e}^{2})$, where I is an identity matrix and ${\hat{\sigma }}_{e}^{2}$ represents the residual variance. The additive genetic covariance between patients (A) was calculated based on their kinship coefficients defining $a\sim N(0,A{\hat{\sigma }}_{a}^{2})$ as well as on the similarity of patients’ SNP genotypes—$u\sim N(0,G{\hat{\sigma }}_{a}^{2}),$ where G is given by
(2)$$G=\frac{MM^{T}}{2\sum_{i=1}^{N_{SNP}^{*}}p_{i}q_{i}},$$
where $M_{ij}\in \left\{2-2p_{i},1-2p_{i},-2p_{i}\right\},$ respectively, stand for homozygous, heterozygous, and alternative homozygous genotypes at *i*th SNP of *j*th individual, $p_{i}/q_{i}$ represent frequency of the reference/alternative allele for *i*th SNP, and $N_{SNP}^{*}$ is a total number of SNPs considered in the model (here, 9,767,423).

In the next step the additive effects of particular SNPs (v) were calculated based on patient SNP additive component (u) using the back-solve method proposed by [76]
(3)$$v={\left({\left(\frac{1-k}{N_{SNP}^{*}}\right)}^{-1}+\frac{1}{k}MA^{-1}M^{T}\right)}^{-1}\frac{1}{k}MA^{-1}\hat{u},$$
where k is a tuning parameter, here k=0.2. The significance of *i*th SNP ($v_{i}$) was assessed by the Wald test: $\frac{v_{i}}{{\hat{\sigma }}_{a}^{2}},$ and the resulting nominal *p*-values were transformed to False Discovery Rates to account for the multiple testing. The above GWAS model was implemented by the GTCA software [76], while the back-solving of SNP effects and hypotheses testing were conducted based on custom-written scripts in R [77] and Python.

### 4.8. Estimation of KEGG Pathway Effects

SNPs were annotated to genes and KEGG pathways using David software [77]. The downstream analysis was carried out separately for RESISTANT and SEVERE phenotypes. In this context of statistical modelling, as compared to GWAS, the goal is to estimate the joint effects of multiple genes (here represented by SNPs) acting functionally together within a metabolic pathway. Individual effects of many of the genes would have been disregarded in a conventional GWAS as nonsignificant, but they may play a role in a joint metabolic response to infection.

The estimation of KEGG pathway effects was based on the following mixed linear model
(4)$$\left|v\right|=\mu +Wt+\varepsilon ,$$
where, $\left|v\right|$ (12,736) is the vector of absolute values of SNP effects estimated from GWAS that were mapped to genes; μ (12,736) is the general mean; t (288) is the random effect KEGG pathway effect with a pre-imposed normal distribution defined by $N\left(0,V\sigma_{t}^{2}\right)$; W (12,736 × 288) is the corresponding incidence matrix for t assigning SNPs to KEGG pathways; ε (12,736) is a vector of residuals distributed as $N\left(0,I\sigma_{\varepsilon }^{2}\right)$. The covariance matrix between KEGG pathways (V) was expressed by the Jaccard similarity coefficient $J\left(i,j\right)=\frac{K}{N}$, where *K* represents the number of genes shared between KEGG *i* and *j,* while *N* represents the total number of genes involved in KEGG *i* and *j*. Variance components were assumed as known, amounting to $\sigma_{t}^{2}=0.3\sigma_{y}^{2}$ and $\sigma_{e}^{2}=1-\sigma_{y}^{2}$. Note that to avoid confounding, within each gene, only one SNP with the highest effect from GWAS was selected for the analysis. The mixed model equations (Henderson C.R. 1984 University of Guelph) were used to obtain solutions for μ and t
(5)$$\left[\begin{array}{c} \hat{\mu }\\ \hat{t} \end{array}\right]={\left[\begin{array}{cc} 1^{T}R^{-1}1 & 1^{T}R^{-1}W\\ WR^{-1}1 & W_{}^{T}R^{-1}W+G_{*}^{-1} \end{array}\right]}^{-1}\left[\begin{array}{c} 1^{T}R^{-1}\left|v\right|\\ W_{}^{T}R^{-1}\left|v\right| \end{array}\right]\;,\;\mathrm{where}\;R=I{\hat{\sigma }}_{\varepsilon }^{2}\;\mathrm{and}\;\;G_{*}=V{\hat{\sigma }}_{t}^{2}.$$

To maximise the computational performance of the estimation ($\hat{\mu }$) and prediction ($\hat{t}$) process, a custom Python program implementing the NumPy library was used [78]. Since all calculations were carried out on a high-performance server, the NumPy library was also used to set the array indexing and ordering, which further improved the computing time as compared to a native Python application. Each element of $\hat{t}$ was assessed for significance ($H_{0}:t_{i}\leq 0\;vs.\;H_{1}:\;t_{i}>0$) by calculating the probability of obtaining a more extreme value using the $N\left(0,\sigma_{t}^{2}\right)$ density function.

## 5. Conclusions

The GWAS studies provide a reliable method to profile patients susceptible to a severe course of COVID-19 by, for instance, the identification of putative genes associated with COVID-19 severity [11,79]. Moreover, the Next-Generation Sequencing (NGS) evaluations provided a full characterization of the entire genome of SARS-CoV-2, and nowadays it is used not only for the discovery of novel molecular variants of this virus, but also to detect novel emerging strains that might pose a threat to public health. The identification of new mutations in the genome of the SARS-CoV-2 virus is also crucial for the development of novel vaccines [80].

Here, we used GWAS analysis to identify metabolic pathways that are associated with a severe COVID-19 infection as well as pathways related to resistance to COVID-19. The main finding of this study were:(1)the Allograft rejection pathway (hsa05330) was significant for the resistance to the COVID-19 infection;(2)27 SNPs marking genes constituting the Allograft rejection pathway, and the majority of these were located on chromosome 6 (19 SNPs), while the remainder were mapped to chromosomes 2, 3, 10, 12, 20, and X.(3)the Allograft rejection pathway comprises several immune system components crucial for the self versus non-self recognition, and also the components of antiviral immunity;(4)no significant metabolic pathway was indicated in the case of susceptibility to COVID-19 and its severe course.

Our study showed that not only are single variants important, but also the cumulative impact of several SNPs within the same pathway seems to matter in the case of severe COVID-19 disease. This evidence supports the notion of a multifactorial background of a disease course. Moreover, as previously indicated by us and others, accumulating evidence suggests inborn errors of the immune system components, especially IFN and TNF superfamily members, are crucial when it comes to the COVID-19 severe course.

## Figures and Tables

**Figure 1 ijms-23-06272-f001:**
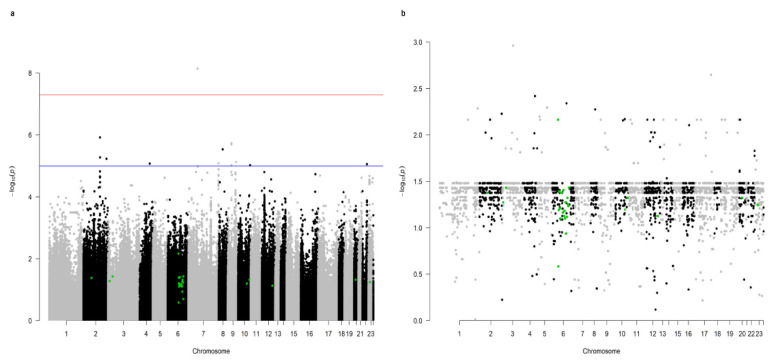
Manhattan plot (**a**) for 9,767,423 SNPs analysed in GWAS. SNPs significant in the Allograft rejection pathway (hsa05330) are marked by green dots. The inside plot (**b**) shows the subset of 12,736 genic SNPs used for the prediction of KEGG pathway effects. The red line indicates the threshold for genome-wide significance (*p* < 5 × 10^−8^) and the blue line for suggestive associations (*p* < 1 × 10^−5^).

**Figure 2 ijms-23-06272-f002:**
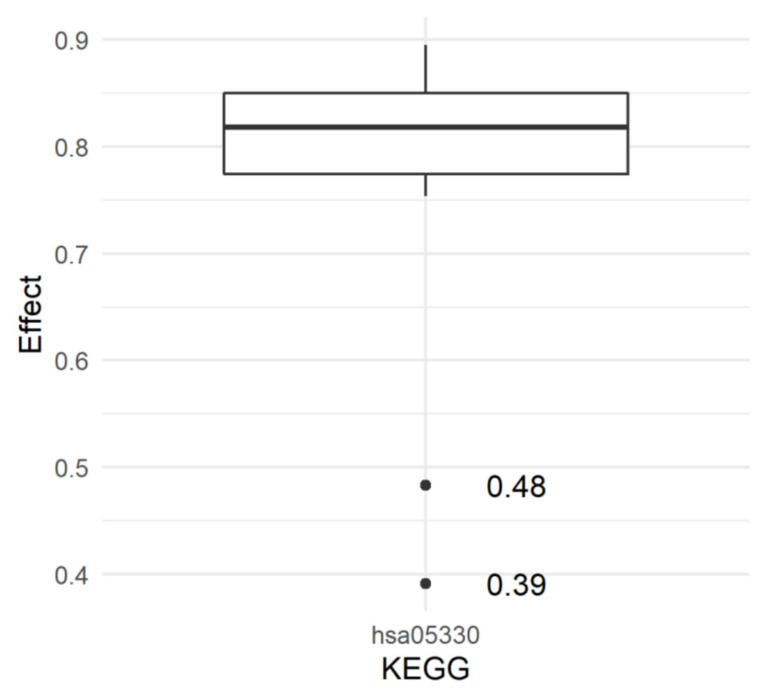
Additive genetic effects of 27 SNPs constituting hsa05330 KEGG pathway on the resistance to COVID-19 infection.

**Figure 3 ijms-23-06272-f003:**
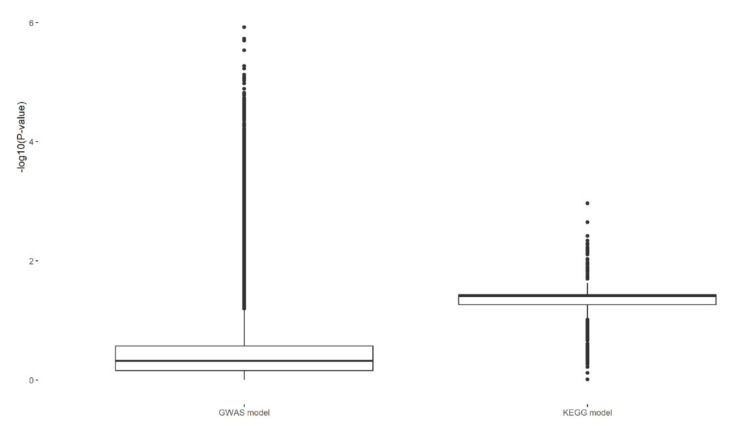
SNP significance of all 9,767,423 SNPs (GWAS model) and the subset of 12,736 (KEGG model).

**Figure 4 ijms-23-06272-f004:**
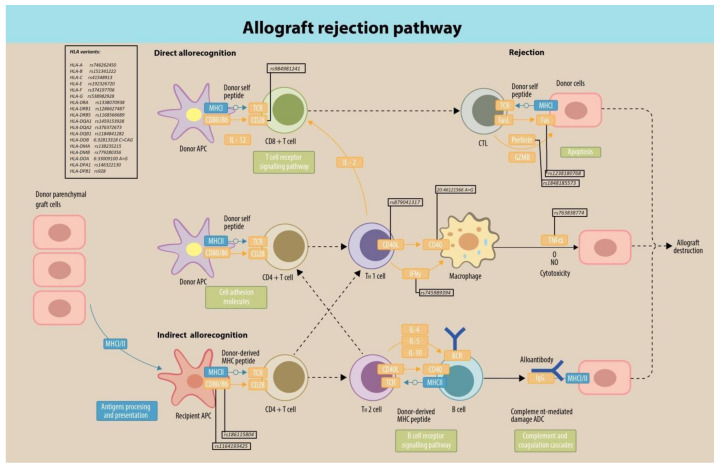
The Allograft rejection pathway (Figure adapted from the KEGG database. Copyright: Paula Dobosz & Wojciech Górski 2022.) with the most significant SNPs from each gene indicated. If the molecule appears on the figure more than once, relevant SNPs have been depicted only once to maintain the clarity of the scheme. Novel SNPs are indicated by their position in bp.

**Table 1 ijms-23-06272-t001:** SNPs representing genes within the Allograft rejection pathway (hsa05330).

Chromosome	Position (bp)	Gene ID	Gene Name	Mutation	Genomic Annotation	SNP ID	Median DP	SD
2	203709432	*ENSG00000178562*	*CD28*	G>A	intron	rs984981241	28	4.145465441
3	119551547	*ENSG00000121594*	*CD80*	G>A	intron	novel	26	4.063210675
3	122057921	*ENSG00000114013*	*CD86*	G>A	intron	rs186115804	30	5.143266314
6	29725272	*ENSG00000204642*	*HLA-F*	C>T	exon	rs374197706	33	6.119863999
6	29827086	*ENSG00000204632*	*HLA-G*	T>C	intron	rs538982928	33	6.636027363
6	29945505	*ENSG00000206503*	*HLA-A*	C>T	3′UTR	rs746262450	32	10.17326727
6	30493031	*ENSG00000204592*	*HLA-E*	T>C	3′UTR	rs192326720	29	5.059734473
6	31271736	*ENSG00000204525*	*HLA-C*	C>T	exon	rs41548913	39	8.517438111
6	31356411	*ENSG00000234745*	*HLA-B*	G>A	exon	rs151341222	43	9.575215658
6	31576590	*ENSG00000232810*	*TNFa*	C>T	intron	rs763838774	28	4.933491701
6	32441500	*ENSG00000204287*	*HLA-DRA*	T>G	intron	rs1338070938	35	6.726123187
6	32528966	*ENSG00000198502*	*HLA-DRB5*	C>T	intron	rs1168566689	39	12.87413207
6	32583693	*ENSG00000196126*	*HLA-DRB1*	T>TC	intron	novel	31	8.339663173
6	32643698	*ENSG00000196735*	*HLA-DQA1*	A>G	Exon of a non-coding transcript	rs1459153928	36	7.155148333
6	32664633	*ENSG00000179344*	*HLA-DQB1*	C>T	intron	rs1184841282	36	7.686703581
6	32746600	*ENSG00000237541*	*HLA-DQA2*	GAGA>G	3′UTR	novel	29	5.463038801
6	32813318	*ENSG00000241106*	*HLA-DOB*	CAG>C	intron	novel	25	4.576624558
6	32935042	*ENSG00000242574*	*HLA-DMB*	G>A	intron	rs779280356	25	4.040525561
6	32964115	*ENSG00000204257*	*HLA-DMA*	T>C	intron	novel	27	4.697217326
6	33009100	*ENSG00000204252*	*HLA-DOA*	G>A	intron	Novel	33	5.700484494
6	33078874	*ENSG00000231389*	*HLA-DPA1*	G>A	intron	rs146322130	28	5.622143954
6	33086775	*ENSG00000223865*	*HLA-DPB1*	CTGTT>C	3′UTR	novel	31	8.032764829
10	70599302	*ENSG00000180644*	*PRF1*	G>A	intron	novel	31	4.807858254
10	88956063	*ENSG00000026103*	*FAS*	T>C	intron	novel	28	4.145442918
12	68156786	*ENSG00000111537*	*IFNG*	C>T	intron	rs745989394	28	4.229532615
20	46121566	*ENSG00000101017*	*CD40*	G>A	intron	novel	26	3.808906036
X	136659930	*ENSG00000102245*	*CD40L*	C>T	3′UTR	rs879041317	12	6.017283757

**Table 2 ijms-23-06272-t002:** Cluster of differentiation proteins with significant impact on the allograft rejection pathway, identified as being significant in the current project.

Cluster of Differentiation Protein	Function	References
CD28	is one of the proteins expressed on T cells, providing costimulatory signals required for T cell activation and survival; provides a potent signal for the production of various interleukins, especially IL-6; molecules CD80 and CD86 are its ligands; the activity of CD80–CD28 complex stimulates the activation of transcription factors NF-κB, promoting IL-2 production	[25,26,27,28,29]
CD40	is a costimulatory protein, a member of the TNF superfamily, constitutively expressed on B cells and antigen-presenting cells; CD40 binds its ligand CD40L, which is transiently expressed on T cells and other non-immune cells under inflammatory conditions; essential in mediating a broad variety of immune and inflammatory responses including T cell-dependent immunoglobulin class switching, germinal centre formation memory B cell development, to name just a few	[22,23]
CD80	is an immunoglobulin, also a ligand for cytotoxic T-lymphocyte antigen 4 (CTLA-4, also known as CD152), which remains constitutively expressed on most of the T cells; present at APCs and their receptors present on the T cells; present specifically on dendritic cells, activated B cells, and macrophages, but also T cells; malfunctioning CD80 molecules are also involved in some pathological conditions, such as lupus erythematosus	[22,25,26,27,28,29]
CD86	is a costimulatory protein, immunoglobulin, constitutively expressed on dendritic cells, pancreatic Langerhans cells, macrophages, B cells (including memory B cells), and on other antigen-presenting cells; provides costimulatory signals crucial for T cell activation and survival; it is also associated with myocarditis and gallbladder squamous cell carcinoma	[22,25,26,27,28,29]
CD152	also known as CTLA-4 (cytotoxic T-lymphocyte-associated protein); it is a receptor that functions as an immune checkpoint protein and downregulates immune responses; it is constitutively expressed on regulatory T cells but found to be upregulated in conventional T cells after activation, being a phenomenon particularly significant in cancers, thus, being important as a background of immunotherapy utilising checkpoint inhibitors	[25,27,29]

## Data Availability

The dataset is freely available for researchers here https://1000polishgenomes.com/ (accessed on 4 April 2022) on a reasonable request (https://www.mdpi.com/1422-0067/23/9/4532 (accessed on 4 April 2022)).

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
