# Peer review of "Beyond GWAS—Could Genetic Differentiation within the Allograft Rejection Pathway Shape Natural Immunity to COVID-19?"

_ijms, 2022, doi:10.3390/ijms23116272_

Round 1

Reviewer 1 Report

The manuscript is very interesting, however, it cannot be completely evaluated due to errors in the Results section. Please address the following comments:

1) The first part of the Results section is missing. It is not clear how much data the authors do not correctly report. Please address this critical issue for a correct evaluation of the entire study;
2) In the discussion section the authors mentioned the effect of checkpoints also targeted in cancer patients. In the Discussion section, please describe the complex relationship existing between immune system, cancer and covid-19 and the possibility of continuing immune checkpoint inhibitors in COVID-19 positive cancer patients as well as the use of immunotherapy for the treatment of severe COVID-19 infection. For this purpose, please see:
- PMID: 33491759
- PMID: 35064248
- PMID: 32785162
- PMID: 32272396
- PMID: 34431319
3) Please add a Table reporting all the relevant socio-demographic and clinical-pathological features of the study participants;
4) In the Introduction or Discussion sections, the authors should add a brief paragraph about the potentialities and application of GWAS studies and NGS evaluations useful for the genomic profiling of susceptible patients or for the identification of virus variants. For this purpose, please see:
- PMID: 32558485
- PMID: 33846767
- PMID: 34759959
5) It is very difficult to understand the main results of this study, the authors should concisely report their main findings. A bullet-point list in the Conclusive remarks should be considered;
6) Some parts of the Discussion section are redundant and too verbose. The entire chapter should be significantly shortened.

Author Response

Reviewer 1

The manuscript is very interesting, however, it cannot be completely evaluated due to errors in the Results section. Please address the following comments:

1) The first part of the Results section is missing. It is not clear how much data the authors do not correctly report. Please address this critical issue for a correct evaluation of the entire study;

Answer: In the revised version of the manuscript, dimensions for each model element were added in order to indicate how much data were used in each model (lines 168-174 and 200-203).

2) In the discussion section the authors mentioned the effect of checkpoints also targeted in cancer patients. In the Discussion section, please describe the complex relationship existing between immune system, cancer and covid-19 and the possibility of continuing immune checkpoint inhibitors in COVID-19 positive cancer patients as well as the use of immunotherapy for the treatment of severe COVID-19 infection. For this purpose, please see:
- PMID: 33491759
- PMID: 35064248
- PMID: 32785162
- PMID: 32272396
- PMID: 34431319

Answer: Done according to the Reviewer’s suggestions (lines 331-395).

3) Please add a Table reporting all the relevant socio-demographic and clinical-pathological features of the study participants;

Answer: Detailed information about the cohort, including demographic and clinical features, can be found in our previous paper by Kaja et al. 2022 (https://www.mdpi.com/1422-0067/23/9/4532). Now this information is stated in the text (lines 105-106).

4) In the Introduction or Discussion sections, the authors should add a brief paragraph about the potentialities and application of GWAS studies and NGS evaluations useful for the genomic profiling of susceptible patients or for the identification of virus variants. For this purpose, please see:
- PMID: 32558485
- PMID: 33846767
- PMID: 34759959

Answer: Done according to the Reviewer’s suggestions (lines 405-411).

5) It is very difficult to understand the main results of this study, the authors should concisely report their main findings. A bullet-point list in the Conclusive remarks should be considered;

Answer: Done according to the Reviewer’s suggestions (lines 412-423). Moreover, the additional Table 2 was added to make Discussion more clear (lines 293-294 and 308-311).

6) Some parts of the Discussion section are redundant and too verbose. The entire chapter should be significantly shortened.

Answer: Done according to the Reviewer’s suggestion. Now, the Discussion section was significantly shortened and rephrased. The additional Table 2 was added to make Discussion more clear (lines 293-294 and 308-311).

Reviewer 2 Report

This manuscript aims to investigate genetic analysis related to metabolic pathways that may be associated with differences in susceptibility and progression to COVID-19. Authors will have to describe better the Results of this manuscript, which currently are not clear. Authors need to improve the Discussion section. The first 2 paragraphs are a general reflection on the human immune system, which may be better suited (if reduced) for the Introduction.

Author Response

Reviewer 2

This manuscript aims to investigate genetic analysis related to metabolic pathways that may be associated with differences in susceptibility and progression to COVID-19. Authors will have to describe better the Results of this manuscript, which currently are not clear. Authors need to improve the Discussion section. The first 2 paragraphs are a general reflection on the human immune system, which may be better suited (if reduced) for the Introduction.

Answer: The Results section was improved to present results of this study more clearly. The Discussion section was substantially rephrased and shortened, and additional relevant data was added. The paragraphs indicated by the Reviewer at the beginning of the Discussion were shortened and unnecessary information was removed. The additional Table 2 was added to make Discussion more clear (lines 293-294 and 308-311).